# IMPLICIT FUNCTIONAL BAYESIAN DEEP LEARNING

## ABSTRACT

Bayesian deep learning (BDL) is believed to be an effective approach to enabling uncertainty estimation and improving the generalisation and robustness of classical deep learning with the help of the Bayesian principle. Considering its non-meaningful weight-space prior and problematic Kullback-Leibler (KL) divergence, functional inference with Wasserstein distance has recently emerged as a promising direction in this field. However, existing efforts require different types of degenerations to achieve tractable Wasserstein distance computation, which limits the predictive and uncertainty estimation capabilities. In this paper, we propose two novel implicit functional BDL (ifBDL) approaches, i.e., *implicit functional Bayesian neural networks* and *implicit functional Bayesian deep ensemble*. The common idea is to implicitly transform the BDL posterior to a Gaussian process via the neural tangent kernel to facilitate tractable 2-Wasserstein distance computation and preserve the neural network parameterization. The experimental evaluations on standard tasks show that ifBDL has superior predictive and uncertainty estimation capabilities compared to existing weight-space and function-space approaches.

## 1 INTRODUCTION

Bayesian deep learning (BDL) (Papamarkou et al., 2024) is believed to be an effective approach to enabling uncertainty estimation and improving the generalisation and robustness of classical deep learning with the help of the Bayesian principle. Some typical models include Bayesian neural networks (BNNs) (Blundell et al., 2015; Gal, 2016), Bayesian deep ensembles (Seligmann et al., 2024), deep kernel processes (Ober et al., 2023), neural processes (Garnelo et al., 2018), and so on. The uncertainty estimation is essential for some safety-critical applications like medical diagnosis (Dolezal et al., 2022) and autonomous driving (Tai et al., 2019), and stronger generalisation and robustness are the keys to applying the trained model in practical scenarios featuring possible distribution shifts and incomplete and noisy data. Recent works have also shown its capability of extending large deep networks like Transformers (Chen & Li, 2023) and GPT (Shen et al., 2024) to yield good weight-uncertainty for calibration, and model averaging.

Similar to all Bayesian approaches, there are two main steps of BDL: prior selection and posterior inference. For prior selection, a straightforward and popular choice is the independent identically distributed (i.i.d) prior for deep learning weights, such as the i.i.d Gaussian distributions. However, this prior is found problematic because 1) it is hard to encode prior knowledge about the underlying functions through distribution design for the large-scale network weights; 2) the samples of such priors over parameters tend to be horizontally linear and lead to pathologies for deep models (Duvenaud et al., 2014; Matthews et al., 2018; Tran et al., 2020); and 3) the effects of the given priors on posterior inference and, consequently, on the resulting distributions over functions are unclear and hard to control due to the complex architecture and non-linearity of the models (Ma & Hernández-Lobato, 2021; Wild et al., 2022). Considering these problems, there is increasing interest in using functional priors instead. A representative example is the Gaussian process (GP) (Rasmussen & Williams, 2005), which can easily encode prior knowledge of underlying functions through kernel design. We call the BDL under functional prior *functional BDL (fBDL)*.

The posterior inference for fBDL is more challenging than weight-space BDL due to the infinite-dimensional functional prior and posterior. The (generalized) variational inference (Knoblauch et al., 2022) of BDL involves a term to minimize the distance between the posterior of BDL and the prior. However, the posterior is a complex distribution over functions without an explicit probability den-

sity function, making the distance measurement to GP prior a challenge. Existing approaches include using a spectral Stein gradient estimator to evaluate the divergence between a BNN and a GP (Sun et al., 2019), as well as using first-order Taylor expansion to approximate the posterior of BDL (Rudner et al., 2022). Both methods are designed for the Kullback-Leibler (KL) divergence, which presents challenges because i) there do not exist density functions with respect to the Lebesgue measure for stochastic processes prior and posterior (Hunt et al., 1992), and ii) the functional KL divergence is not always well-defined (Burt et al., 2020) as the variational posterior measure should be absolutely continuous with respect to the prior measure to guarantee the exist of Radon-Nikodym derivative; otherwise, the KL divergence can be infinite (Gray, 2011; Matthews et al., 2016). Consequently, efforts have been made to use the Wasserstein distance instead. For instance, the dual form of the 1-Wasserstein distance is employed to evaluate the distance between a BNN and a GP (Tran et al., 2022; Wu et al., 2024). Compared to the 1-Wasserstein distance, which requires a number of function samples for approximation, the 2-Wasserstein distance can be much more easily evaluated because it has a closed form for Gaussian measures on a given measurement set. To utilize the 2-Wasserstein distance, another GP using a deep neural network as the mean function approximates the posterior of BDL, and then the 2-Wasserstein distance is used to evaluate the distance between a BNN and a GP (Wild et al., 2022). However, this GP degeneration reduces the uncertainty modelling and model generalization capability of the original BDL, as verified in the experiments.

In this paper, we propose a novel functional BDL using the 2-Wasserstein distance via the neural tangent kernel. The NTK (Jacot et al., 2018; Lee et al., 2019) captures the evolution of deep neural networks under gradient descent. Under certain conditions, the distribution of functions learned by such networks can be well approximated by a GP parameterized by the NTK. Motivated by these properties of the NTK, we establish a connection between the BDL posterior and the NTK-based GP transformation, resulting in our novel functional BDL method. Experimental evaluations demonstrate that this approach outperforms the approximation using a GP with the deep neural network as the mean function. Our main contributions are summarised as follows:

- We propose to use the 2-Wasserstein distance with good statistical properties instead of KL divergence to achieve functional BDL. This approach can avoid many limitations associated with using KL divergence in infinite-dimensional function space;

- A neural tangent kernel-based approach is employed to transform a BDL posterior to a GP. This transformation preserves more uncertainty estimation and generalization capability of BDL compared to the approximation using GP with deep neural networks as a mean function.

- We introduce and verify the effectiveness of two functional BDL approaches, i.e., *functional Bayesian neural networks* and *functional Bayesian deep ensembles*. These approaches are evaluated through comparative experiments on standard tasks with several existing parameter space and functional space variational inference methods.

## 2 BACKGROUND

### 2.1 BAYESIAN DEEP LEARNING

Consider a dataset $\mathcal{D} = \{\mathbf{X}, \mathbf{Y}\} = \{(x_i, y_i)\}_{i=1}^n$, where $x \in \mathbb{R}^d$ represents $d$-dimensional input and $y \in \mathbb{R}^c$ represents $c$-dimensional target. Classical (determinate) deep learning (DL) aims to build a function mapping $f(y|x; \mathbf{w})$ between input and target weighted by parameter $\mathbf{w}$, while Bayesian deep learning (BDL) aims to learn a distribution over all possible functions $p(f|\mathcal{D})$ instead of a single $f$. Such distribution over functions is able to improve the model generalization and uncertainty modelling (Papamarkou et al., 2024). In this paper, we use two examples of BDL: Bayesian neural networks and Bayesian deep ensembles.

### 2.1.1 BAYESIAN NEURAL NETWORKS

A Bayesian neural network (BNN) assigns prior $p_0(\mathbf{w})$ and likelihood $p(\mathcal{D}|\mathbf{w})$ to neural network weights. According to Bayes' theorem, the posterior distribution of $\mathbf{w}$ can be calculated as $p(\mathbf{w}|\mathcal{D}) \propto p(\mathcal{D}|\mathbf{w})p_0(\mathbf{w})$. Given test data $x^*$, the predictive distribution can be obtained by $p(y^*|x^*, \mathcal{D}) = \mathbb{E}_{p(\mathbf{w}|\mathcal{D})}(p(y^*|x^*, \mathbf{w})$. It is important to note that the distribution over functions

$p(f|\mathcal{D})$ is induced by the posterior distribution $p(\mathbf{w}|\mathcal{D})$. Since the posterior is intractable for practical neural network architectures, the variational posterior inference is used to fit a tractable approximation posterior $q_{\boldsymbol{\theta}}(\mathbf{w})$, parameterized by $\boldsymbol{\theta}$, to the exact posterior $p(\mathbf{w}|\mathcal{D})$. This is achieved by minimizing their KL divergence, which is equivalent to maximizing the evidence lower bound (ELBO):

$$\mathcal{L}_{q_{\boldsymbol{\theta}}(\mathbf{w})} = \mathbb{E}_{q_{\boldsymbol{\theta}}(\mathbf{w})}[\log p(\mathcal{D} \mid \mathbf{w})] - \mathcal{KL}[q_{\boldsymbol{\theta}}(\mathbf{w})\|p_0(\mathbf{w})]. \tag{1}$$

Bayes By Backprop (BBB), proposed by Blundell et al. (2015), is one of the most used variational inference learning algorithms for BNNs in weight space. BBB leverages a fully factorized Gaussian assumption about variational posterior and employs the reparameterization trick (Kingma & Welling, 2014) to obtain unbiased gradient estimates of ELBO with respect to model parameters. Several variants of BBB have been proposed in the field (Gal & Ghahramani, 2016; Marino et al., 2018; Santana & Hernández-Lobato, 2022). Despite their differences, these variants share a similar basic framework. However, the challenge remains in choosing an appropriate prior for network weights and understanding its impact on inference and the resulting posterior distribution.

### 2.1.2 BAYESIAN DEEP ENSEMBLE

Deep ensemble (Lakshminarayanan et al., 2017) trains multiple ($M$) independent determinate neural networks $\{f_i(y|x, \mathbf{w}_i)\}_i^M$ with different initializations. The posterior of weights can be considered as in the form of a mixture of deltas $q(\mathbf{w}) = \frac{1}{M}\sum_i^M \delta(\mathbf{w} - \mathbf{w}_i)$ (Deng et al., 2022). Given test data $x^*$, the predictive distribution can be obtained as $p(y^*|x^*, \mathcal{D}) = \frac{1}{M}\sum_i^M f_i(y^*|x^*)$. Deep ensembles are effective for uncertainty modelling and out-of-distribution predictions. Their success is attributed to their relation to Bayesian inference and posterior approximation (Wilson & Izmailov, 2020; He et al., 2020). When we assign a prior $p_0(\mathbf{w})$ to the deep ensemble, we can derive a similar (generalized) variational inference objective for the deep ensemble by adding a distance regularizer on $\mathbf{w}$, similar to BNNs,

$$\mathcal{L}_{\{\mathbf{w}_i\}} = \sum_i^M \log p(\mathcal{D} \mid \mathbf{w}_i)] - \mathcal{KL}[q(\mathbf{w})\|p_0(\mathbf{w})]. \tag{2}$$

We refer to the deep ensemble with a given prior as *Bayesian deep ensemble (BDE)* in this paper. The distribution over functions $p(f|\mathcal{D})$ is also induced by the posterior distribution $p(\mathbf{w}|\mathcal{D})$.

### 2.2 FUNCTIONAL POSTERIOR INFERENCE

Suppose $p_0(f)$ is a functional prior for BNNs defined on a probability space $(\Omega, \mathcal{F}, P)$ (a separable metric and complete Polish space) with a stochastic process $f(\cdot) : \mathcal{T} \mapsto \mathcal{R}$, where $\mathcal{T}$ is an infinite compact index set for $f(\cdot)$. Similar to the idea of Bayesian inference in parameter space, $p_0(f)$ combined with the likelihood $p(\mathcal{D}|f)$ yields the posterior $p(f|\mathcal{D})$ given observed data. However, in most cases, this posterior cannot be solved analytically. Performing variational inference directly in function space aims to find an approximate posterior $q(f)$ on $(\Omega, \mathcal{F}, P)$ that approximates the truly posterior. This is typically achieved by minimizing the KL divergence between them. The KL divergence between the distributions in function space is defined based on their Radon-Nikodym derivatives (Gray, 2011)

$$\mathcal{KL}[q(f)\|p(f)] = \int_{\Omega} \log \left\{ \frac{\mathrm{dq}}{\mathrm{dp}}(f) \right\} dq(f) \tag{3}$$

where $\frac{\mathrm{dq}}{\mathrm{dp}}(f)$ is the Radon-Nikodym derivative between $q(f)$ and $p(f)$ under the condition that $q(f)$ is absolutely continuous with respect to $p(f)$. It is important to note that $p(f)$ and $q(f)$ are actually probability measures, as there are no densities with respect to the Lebesgue measure in infinite-dimensional function space (our notation is slightly garbled here). According to the measure-theoretic definition of Bayes' theorem (Schervish, 2012), the Radon-Nikodym derivative of the posterior $p(f|\mathcal{D})$ w.r.t. the prior $p_0(f)$ is defined as

$$\frac{\mathrm{dp}(f|\mathcal{D})}{\mathrm{dp}_0(f)} = \frac{p(\mathcal{D}|f)}{p(\mathcal{D})} \tag{4}$$

where $p(\mathcal{D}) = \int p(\mathcal{D}|f)\mathrm{dp}(f)$ represents the marginal likelihood. By applying the chain rule of Radon-Nikodym derivatives, we obtain the KL divergence between $q(f)$ and $p(f|\mathcal{D})$ in function

space:

$$\mathcal{KL}[q(f)\|p(f|\mathcal{D})] = \int \log\left\{\frac{dq}{dp}(f)\right\} dq(f) - \int \log\left\{\frac{dp(f|\mathcal{D})}{dp(f)}\right\} dq(f) \tag{5}$$
$$= \mathcal{KL}[q(f)\|p(f)] - \mathbb{E}_{q(f)}[\log p(\mathcal{D}|f)] + \log p(\mathcal{D}).$$

Consequently, $q(f)$ can be equivalently optimized by maximizing the functional evidence lower bound (fELBO)

$$\mathcal{L}_{q(f)} := \mathbb{E}_{q(f)}\left[\log p(\mathcal{D} \mid f)\right] - \mathcal{KL}[q(f)\|p_0(f)]. \tag{6}$$

The key challenge lies in effectively estimating the KL divergence between the stochastic process prior $p_0(f)$ and the variational posterior $q(f)$. Although such KL divergence can be approximated in some ways (Sun et al., 2019), its practical validity may be limited because its application strictly assumes the existence of Radon-Nikodym derivatives between $q(f)$ and $p_0(f)$. In cases where, for example, the prior and the variational posterior correspond to two neural networks with different structures, we encounter situations where $\mathcal{KL}[q(f)\|p_0(f)] = \infty$ (Burt et al., 2020).

In contrast to KL divergence, Wasserstein distance (Kantorovich, 1960; Villani, 2021) is a rigorously defined distance metric on probability measures satisfying non-negativity, symmetry and the triangular inequality (Panaretos & Zemel, 2019). Originally proposed for the optimal transport problem, Wasserstein distance has gained popularity in the machine learning community in recent years (Arjovsky et al., 2017). Let us consider a Polish space $(\mathcal{P}, \|\cdot\|)$, where the $\tau$-Wasserstein distance between probability measures $\mu$ and $\nu$ in $(\mathcal{P}, \|\cdot\|)$ is defined as

$$\mathcal{W}_\tau(\mu, \nu) = \left(\inf_{\gamma \in \Gamma(\mu,\nu)} \int_{\mathcal{P}\times\mathcal{P}} \|x-y\|^\tau \, \mathrm{d}\gamma(x,y)\right)^{1/\tau} \tag{7}$$

where $\Gamma(\mu,\nu)$ represents the set of joint measures or couplings $\gamma$ with marginals $\mu$ and $\nu$ on $\mathcal{P} \times \mathcal{P}$. The term $\|\cdot\|^\tau$ quantifies the effort required to transport one unit mass from measure $\mu$ to $\nu$, and the $\tau$-Wasserstein distance measures the minimal cost of reconfiguring the mass distribution of one probability measure to match another. Tran et al. (2022) proposed using the Kantorovich-Rubinstein dual form of the 1-Wasserstein distance (Villani et al., 2009; Arjovsky et al., 2017) to evaluate the distance between a BNN and a GP,

$$\mathcal{W}_1[q(f)\|p_0(f)] = \max_{\|\phi\|_{\leq 1}} \mathbb{E}_{q(f)}[\phi(f)] - \mathbb{E}_{p_0(f)}[\phi(f)] \tag{8}$$

where $\|\phi\|_{\leq 1}$ indicates that $\phi$ is constrained to be a 1-Lipschitz function. Unlike the 1-Wasserstein distance, which requires function samples for approximation, the 2-Wassessetin distance can be more easily evaluated because it has a closed form for Gaussian measures on a given measurement set (Gelbrich, 1990). To use 2-Wasserstein distance, Wild et al. (2022) proposed to use another GP with the deep neural network as the mean function to approximate the posterior on measurement set $\mathbf{X}$ (named GWI), $q(f_\mathbf{X}) \approx \mathcal{N}(m_{\mathbf{X},q}, \Sigma_{\mathbf{X},q})$,

$$\mathcal{W}_2[q(f_\mathbf{X})\|p_0(f_\mathbf{X})] = \|m_{\mathbf{X},q} - m_{\mathbf{X},0}\|_2^2 + tr(\Sigma_{\mathbf{X},0}) + tr(\Sigma_{\mathbf{X},q}) - 2tr([\Sigma_{\mathbf{X},q}^{1/2}\Sigma_{\mathbf{X},0}\Sigma_{\mathbf{X},q}^{1/2}]^{1/2}) \tag{9}$$

where $p_0(f_\mathbf{X}) = \mathcal{N}(m_{\mathbf{X},0}, \Sigma_{\mathbf{X},0})$ represents GP prior on measurement set $\mathbf{X}$, and $tr()$ denotes the trace of an operator. However, it is important to note that the uncertainty of $f_\mathbf{X}$ is directly associated with $\Sigma_{\mathbf{X},q}$ (usually predefined or jointly parameterized) rather than the variances of all network weights. Consequently, this GP transformation reduces the uncertainty modelling and model generalization capability of BDL.

## 3 PROPOSED METHODS

In this section, we introduce two new functional BDL approaches with 2-Wasserstain distance as the regularizer via neural tangent kernel: functional BNN and functional BDE.

### 3.1 IMPLICIT FUNCTIONAL BAYESIAN NEURAL NETWORKS

We aim to extend the weight-space BNN to a functional BNN by revising its variational inference objective (1) to the following functional counterpart

$$\mathcal{L}_{q(f)} = \mathbb{E}_{q_\theta(\mathbf{w})}[\log p(\mathcal{D} \mid \mathbf{w})] - \mathcal{W}_2[q(f; \mathbf{w})\|p_0(f)] \tag{10}$$

where $q(f; \mathbf{w})$ represents the BNN posterior distribution over functions induced by the distribution $q_\theta(\mathbf{w})$. It is important to note that we cannot directly use (9) because $q(f; \mathbf{w})$ is a BNN posterior rather than GP. Next, we propose a GP transformation of $q(f; \mathbf{w})$ to enable the efficient 2-Wasserstein distance evaluation. Our idea is based on Neural Tangent Kernel (NTK) (Jacot et al., 2018; Arora et al., 2019), which is an important concept linking GP and neural networks. The NTK is defined as

$$\hat{\Theta}_t (\mathbf{x}, \mathbf{x}') = \langle \nabla_{\mathbf{w}} f (\mathbf{x}, \mathbf{w}_t), \nabla_{\mathbf{w}} f (\mathbf{x}', \mathbf{w}_t) \rangle$$

where $f(\mathbf{x}, \mathbf{w}_t)$ is the current output of NNs with parameter $\mathbf{w}_t$ and $\hat{\Theta}_t$ denotes the *empirical NTK*. When the network width and training step go to infinity, such empirical NTK will converge to a positive definite kernel $\Theta$, that is, NTK, which then stays constant during the training.

At a given training step $t$, we can linearize the stochastic function defined by the BNN posterior around the mean value of $\mathbf{w}_t \sim \mathcal{N}(\boldsymbol{\mu}_t, \boldsymbol{\sigma}_t)$ as

$$\tilde{f}_t(\cdot; \mathbf{w}) \approx f(\cdot; \boldsymbol{\mu}_t) + \nabla_{\mathbf{w}} f (\cdot; \boldsymbol{\mu}_t) (\mathbf{w} - \boldsymbol{\mu}_t) \tag{11}$$

where $\nabla_{\mathbf{w}} f (\cdot; \boldsymbol{\mu}_t)$ represents the Jacobian of $f$ at $\boldsymbol{\mu}_t$. We can safely do this linearization because of the 'lazy-training' property of deep neural networks, especially with large network widths (Jacot et al., 2018; Lee et al., 2019). Further, Lee et al. (2019) proved that the linearized neural network $f(\mathbf{x}; \mathbf{w}_t)$ with randomly initialized $f_0(\mathbf{x})$, when $t \to \infty$, satisfies

$$f_\infty^{lin}(\mathbf{x}) := f_\infty(\mathbf{x}) = f_0(\mathbf{x}) - \Theta_{\mathbf{x}\mathbf{X}} \Theta_{\mathbf{X}}^{-1} (f_0(\mathbf{X}) - \mathbf{Y}) \tag{12}$$

where $\Theta_{\mathbf{x}\mathbf{X}} = \Theta(\mathbf{x}, \mathbf{X})$ and $\Theta_{\mathbf{X}\mathbf{X}}^{-1} = \Theta^{-1}(\mathbf{X}, \mathbf{X})$. At step $t$, we know the network function in (11) satisfies GP distribution,

$$\tilde{f}_t \sim \text{GP}(f(\cdot; \boldsymbol{\mu}_t), \Lambda_{\mathbf{x}\mathbf{x}'}), \ \Lambda_{\mathbf{x}\mathbf{x}'} = \nabla_{\mathbf{w}} f (\mathbf{x}; \boldsymbol{\mu}_t) \Sigma_t \nabla_{\mathbf{w}} f (\mathbf{x}'; \boldsymbol{\mu}_t)^\top, \ \Sigma_t = \text{diag}(\boldsymbol{\sigma}_t). \tag{13}$$

If using it as the $f_0$, we can derive the distribution $f_\infty \overset{d}{\sim} GP(m_\infty(\mathbf{x}), k_\infty(\mathbf{x}, \mathbf{x}'))$, where

$$
\begin{aligned}
f_\infty \sim & \text{GP}(m_\infty(\mathbf{x}), k_\infty(\mathbf{x}, \mathbf{x}')) \\
m_\infty(\mathbf{x}) = & f(\cdot; \boldsymbol{\mu}_t) - \Theta(\mathbf{x}, \mathbf{X}) \Theta_{\mathbf{X}\mathbf{X}}^{-1}(f(\mathbf{X}; \boldsymbol{\mu}_t) - \mathbf{Y}) \\
k_\infty(\mathbf{x}, \mathbf{x}') = & \Lambda_{\mathbf{x}\mathbf{x}'} - \Lambda_{\mathbf{x}\mathbf{X}} \Theta_{\mathbf{X}\mathbf{X}}^{-1} \Theta_{\mathbf{X}\mathbf{x}'} - \Theta_{\mathbf{x}\mathbf{X}} \Theta_{\mathbf{X}\mathbf{X}}^{-1} \Lambda_{\mathbf{X}\mathbf{x}'} + \Theta_{\mathbf{x}\mathbf{X}} \Theta_{\mathbf{X}\mathbf{X}}^{-1} \Lambda_{\mathbf{X}\mathbf{X}} \Theta_{\mathbf{X}\mathbf{X}}^{-1} \Theta_{\mathbf{X}\mathbf{x}'}.
\end{aligned}
\tag{14}
$$

The derivation detail is given in the Appendix. As illustrated in Figure 1a, the procedure is: at step $t$, we can firstly evaluate the current implicit empirical NTK at $\mu_t$ by $\hat{\Theta}_{t,\boldsymbol{\mu}_t} (\mathbf{x}, \mathbf{x}') = \langle \nabla_{\mathbf{w}} f (\mathbf{x}; \boldsymbol{\mu}_t), \nabla_{\mathbf{w}} f (\mathbf{x}'; \boldsymbol{\mu}_t) \rangle$ and $\Lambda_{\mathbf{x}\mathbf{x}'} = \nabla_{\mathbf{w}} f (\mathbf{x}; \boldsymbol{\mu}_t) \Sigma_t \nabla_{\mathbf{w}} f (\mathbf{x}'; \boldsymbol{\mu}_t)^\top$; then we transform $q(f; \mathbf{w})$ to a GP as in (14); and finally we calculate the 2-Wasserstein distance $\mathcal{W}_2[\text{GP}(m_\infty(\mathbf{x}), k_\infty(\mathbf{x}, \mathbf{x}')) \| p_0(f)]$ as the regularizer to update the BNN together with the data likelihood.

This regularizer can be understood as follows: At training step t, imagine training the network from its current state to infinity using the standard neural network training procedure (without regularization) with the observed data $\langle \mathbf{X}, \mathbf{Y} \rangle$. The stochastic network will converge to the above GP. The regularizer constrains the BNN posterior update by constraining this GP to remain close to the prior in Wasserstein space. Note that our GP transformation preservers all parameters of BNNs (i.e., $\boldsymbol{\mu}, \boldsymbol{\sigma}$), while the transformation in GWI only preserves the $\boldsymbol{\mu}$ and uses a shared variance for all predictions. Consequently, our NTK-based approach could provide better uncertainty modelling capability than GWI. We call the functional objective (10), with the distance defined by $\mathcal{W}_2[\text{GP}(m_\infty(\mathbf{x}), k_\infty(\mathbf{x}, \mathbf{x}')) \| p_0(f)]$, *implicit functional Bayesian neural networks (ifBNN)*.

One issue with the above procedure is that the 2-Wasserstein distance between two Gaussian processes exists in infinite-dimensional space. Fortunately, we have the following theorem:

**Theorem 1 (Mallasto & Feragen (2017))** *The 2-Wasserstein metric between Gaussian distributions on finite samples converges to the Wasserstein metric between GPs, that is, if $f_{in} \sim \mathcal{N}(m_{in}, \Sigma_{in})$, $f_i \sim GP(m_i, k_i)$ for $i = 1, 2$, then*

$$\lim_{n \to \infty} \mathcal{W}_2^2 (f_{1n}, f_{2n}) = \mathcal{W}_2^2 (f_1, f_2). \tag{15}$$

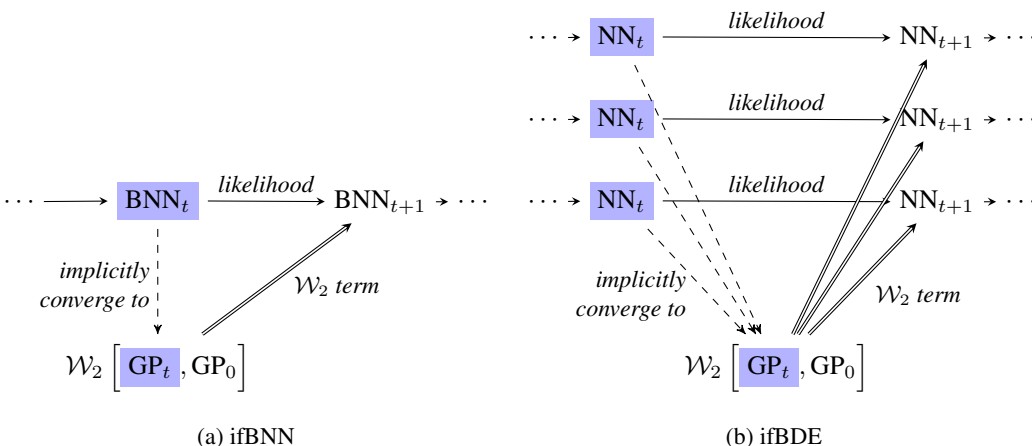

(a) ifBNN                                      (b) ifBDE

Figure 1: Conceptual frameworks of ifBNN and ifBDE

According to the above theorem, we can approximate $\mathcal{W}_2[\text{GP}\left(m_\infty(\mathbf{x}), k_\infty\left(\mathbf{x}, \mathbf{x}'\right)\right)\|p_0(f)]$ via the tractable 2-Wasserstein metric between Gaussian distributions over a finite number of measurement samples. Note that although the KL divergence between two GP processes is also approximated by a measurement set in (Sun et al., 2019), they are different. The limit in (15) from Theorem 1 is relatively easier to achieve compared to the KL divergence because the limit in (15) can be reached by simply increasing the measurement sample size. In contrast, achieving the limit for KL divergence requires searching for the best subset among all possible subsets. The entire process of ifBNN is summarized in Algorithm 1 (in Appendix).

## 3.2 IMPLICIT FUNCTIONAL BAYESIAN DEEP ENSEMBLE

This section aims to extend the weight-space BDE to functional BDE by revising its objective (2) to the following functional counterpart

$$\mathcal{L}_{\{\mathbf{w_i}\}} = \sum_i^M \log p(\mathcal{D} \mid \mathbf{w_i}) - \mathcal{W}_2[q(f; \mathbf{w})\|p_0(f)] \tag{16}$$

where $q(f; \mathbf{w})$ represents functional posterior distribution induced by distribution $q(\mathbf{w})$. Since the prior $p_0$ typically comes from domain expert knowledge about the underlying functions for a target task, it is used to constrain the final aggregated function distribution rather than each ensemble in (16). We call this *implicit functional Bayesian deep ensemble (ifBDE)*. The concept is visually represented in Figure 1b. The next steps involve obtaining a GP transformation of the aggregated ensembles and evaluating the 2-Wasserstein distance with the prior.

Continuing with (12) and for an arbitrarily initialized $f_0 \overset{d}{\sim} \text{GP}(0, k(\mathbf{x}, \mathbf{x}'))$, the posterior is $f_\infty \overset{d}{\sim}$ $\text{GP}\left(m_\infty(\mathbf{x}), k_\infty\left(\mathbf{x}, \mathbf{x}'\right)\right)$, where

$$m_\infty(\mathbf{x}) = \Theta(\mathbf{x}, \mathbf{X})\Theta_\mathbf{X}^{-1}\mathbf{Y}$$
$$k_\infty(\mathbf{x}, \mathbf{x}') = k_{\mathbf{x}\mathbf{x}'} + \Theta(\mathbf{x}, \mathbf{X})\Theta_\mathbf{X}^{-1}k_\mathbf{X}\Theta_\mathbf{X}^{-1}\Theta(\mathbf{X}, \mathbf{x}') - \left(\Theta(\mathbf{x}, \mathbf{X})\Theta_\mathbf{X}^{-1}k_{\mathbf{X}\mathbf{x}} + \Theta(\mathbf{x}, \mathbf{X})\Theta_\mathbf{X}^{-1}k_{\mathbf{X}\mathbf{x}'}\right). \tag{17}$$

We can see that $k_\infty$ could be significantly simplified if $k$ is equal to NTK $\Theta$. This can be achieved by adding a random function to the original network function, $\tilde{f}(\cdot) = f(\cdot) + g(\cdot)$, where $g(\cdot) = \nabla_\mathbf{w} f(\cdot, \mathbf{w})\mathbf{w}^*$ and $\mathbf{w}^* = \text{concat}\left(\{\tilde{\mathbf{w}}^{\leq L}, \mathbf{0}\}\right)$. It has been proven that $g(\cdot) \overset{d}{\to} \text{GP}\left(0, \Theta^{\leq L}\right)$ and $\tilde{f}_0(\cdot) = f_0(\cdot) + g(\cdot) \overset{d}{\to} \text{GP}(0, \Theta)$ under NTK parameterization He et al. (2020). Then the covariance matrix of $\tilde{f}_\infty(\cdot)$ becomes

$$\tilde{k}\infty(\mathbf{x}, \mathbf{x}) = \Theta(\mathbf{x}, \mathbf{x}) - \Theta(\mathbf{x}, \mathbf{X})\Theta^{-1}\mathbf{X}\Theta(\mathbf{X}, \mathbf{x}) \tag{18}$$

and it is interesting to see that this is equal to the GP posterior.

On top of that, we obtain a GP transformation for each neural network in the ensemble, i.e., $\{\mathrm{GP}_i\}_{i=1:M}$, and then a straightforward idea to aggregate them is to use the *average* below:

$$\mathrm{GP}_t = \frac{1}{M}\sum_i^M \mathrm{GP}_{i,t},\ m_t(x) = \frac{1}{M}\sum_i^M m_{i,t}(x),\ k_t(x,x') = \frac{1}{M}\sum_i^M k_{i,t}(x,x'). \tag{19}$$

where

$$\tilde{f}_i(\mathbf{x}) \sim \mathrm{GP}_i(m_i(\mathbf{x}), k_i(\mathbf{x},\mathbf{x}')),\ m_i(\mathbf{x}) = \Theta_{\mathbf{xX}}\Theta_{\mathbf{XX}}^{-1}\mathbf{Y},\ k_i(\mathbf{x},\mathbf{x}') = \Theta_{\mathbf{xx}'} - \Theta_{\mathbf{xX}}\Theta_{\mathbf{XX}}^{-1}\Theta_{\mathbf{Xx}}. \tag{20}$$

However, this straightforward average overlooks the underlying geometry of probability distributions. To align with the Wasserstein distance regularization used, we propose using the Wasserstein Barycenter (Stromme, 2020) of GPs, which is defined in a Wasserstein geometry space that captures the underlying structure of probability distributions.

**Definition 1 (Wasserstein Barycenter (Cuturi & Doucet, 2014))** *A Wasserstein barycenter of $M$ measures $\nu_1, \ldots, \nu_M$ in $\mathbb{P} \in P(\Omega)$ is $\arg\min_\mu \sum_i^M \xi_i \mathcal{W}_p(\mu, \nu_i)$, where $\xi_i > 0, \sum \xi_i = 1$.*

For our problem, we assume no preference over ensemble members, so barycentric coordinates are fixed as $\xi_i = \frac{1}{M}$ and $p = 2$. When $\{\nu_i = \mathcal{N}(m_i, \Sigma_i)\}$ are finite-dimensional Gaussian distributions, the mean of their Wasserstein barycenter is simply $\frac{1}{M}\sum_i m_i$, but there is no explicit analytic form for the covariance matrix. Fortunately, there is an efficient and differentiable fixed-point method to calculate it as follows.

**Theorem 2 ((Álvarez-Esteban et al., 2016))** *Assume $\Sigma_1, \ldots, \Sigma_M \in \mathbb{R}^{d \times d}$ are symmetric positive semidefinite matrices, with at least one of them positive definite. Consider some symmetric positive semidefinite matrices $S_0$ and define*

$$S_{n+1} = S_n^{-1/2}\left(\sum_{j=1}^k (S_n^{1/2}\Sigma_j S_n^{1/2})^{1/2}\right)^2 S_n^{-1/2}, n \geq 0. \tag{21}$$

*If $\mathcal{N}(0, \Sigma_0)$ is the barycenter of $\mathcal{N}(0, \Sigma_1), \ldots, \mathcal{N}(0, \Sigma_k)$, then*

$$\mathcal{W}_2(\mathcal{N}(0, S_n), \mathcal{N}(0, \Sigma_0)) \to 0 \tag{22}$$

*as $n \to \infty$.*

According to this theorem, we can obtain the (approximated) covariance matrix through several iterations with a random initialization (we used the Euclidean mean from (19) in the implementation). For $M = 20, d = 50$, this procedure can achieve an accuracy of $10^{-10}$ in fewer than 14 steps. The following theorem further ensures that the barycenter of infinite-dimensional GPs can be approximated by that of their finite-dimensional Gaussian distribution counterparts.

**Theorem 3 ((Mallasto & Feragen, 2017))** *Assuming the barycenter of a population of GPs is non-degenerate, the barycenter of the finite-dimensional restrictions converges to the barycenter of GPs.*

With the above theoretical guarantee, we first find the Wasserstein barycenter of all member networks via

$$\mathrm{GP}_t = \arg\min \sum_i^M \mathcal{W}_2\left[\mathrm{GP}_i - \mathrm{GP}\left(0, \hat{\Theta}_{\mathbf{w}_i}^{\leq L}\right) \| \mathrm{GP}_t\right] \tag{23}$$

where a GP component is added to reduce the effect from an additional $g$ function for each ensemble. A measurement set is sampled and used to transform the above infinite-dimensional problem into a finite-dimensional one. Then, the fixed-point algorithm in (21) is used to obtain the barycenter covariance. Together with the weighted mean, the Wasserstein distance $\mathcal{W}_2[\mathrm{GP}_t \| p_0(f)]$ is used in (16), which is also in the finite-dimensional space due to the finite-dimensional nature of $\mathrm{GP}_t$ but is guaranteed to converge to its infinite-dimensional counterpart as the measurement set size approaches infinity. Note that to ensure consistency, the sampled measurement set will be used for both barycenter computation and the final prior regularization. The whole process is summarized in Algorithm 2 (in Appendix).

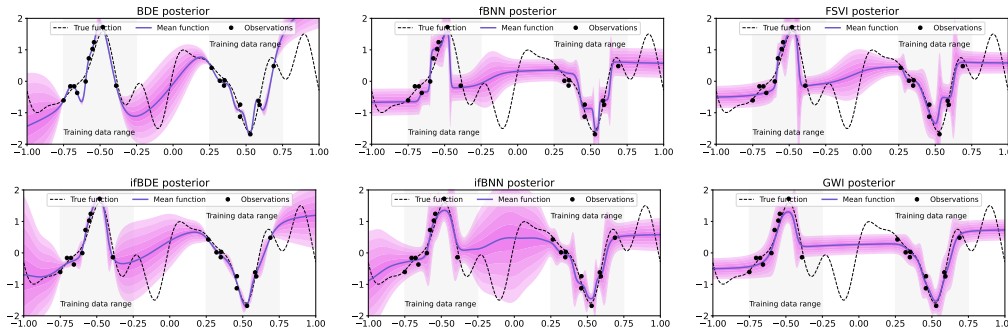

Figure 2: Extrapolation Illustrative Examples. The dashed black line is the ground true function and black dots denote 20 observations. The blue line corresponds to the mean of approximate posterior predictions and shadow areas represent the predictive standard deviations.

## 4 RELATED WORK

Due to the limitations of parameter-space variational inference, such as the intractability of specifying meaningful priors, Sun et al. (2019) proposed a functional ELBO to match a GP prior and the variational posterior over functions for BNNs via a spectral Stein gradient estimator designed for implicit distributions (Shi et al., 2018). Concurrently, Wang et al. (2019) proposed a particle optimization variational inference method in function spaces for posterior approximation in BNNs. Rudner et al. (2020; 2022) pointed out that the supremum of marginal KL divergence over finite measurement sets cannot be solved analytically for the estimation of functional KL divergence. They proposed approximating the distributions over functions as Gaussian via the linearization of their mean parameters, deriving a tractable and well-defined variational objective since the functional prior and variational posterior are two BNNs that share the same network structures. Ma & Hernández-Lobato (2021) randomized the number of finite measurement points to derive an alternative grid-functional KL divergence, which can avoid some limitations of KL divergence between stochastic processes. However, all these methods are based on KL divergence between stochastic processes, which might be ill-defined for a wide class of distributions and lead to an invalid variational objective (Burt et al., 2020).

Considering the potential weaknesses of KL divergence, some recent works have explored using Wasserstein distance (Kantorovich, 1960; Villani, 2021) as a replacement. Tran et al. (2020) proposed matching a BNN prior to a GP prior by minimizing the 1-Wasserstein distance to obtain more interpretable functional priors in BNNs. However, they used stochastic gradient Hamiltonian Monte Carlo (SGHMC) rather than variational inference to approximate the posterior. Wild et al. (2022) developed a functional variational objective called GWI, where both the functional prior and posterior are Gaussian measures, and the dissimilarity measure is the 2-Wasserstein distance. However, the critical GP degeneration makes it less expressive than the original BDL in terms of uncertainty. Our method retains the original BDL parameterization and facilitates tractable 2-Wasserstein distance regularization.

## 5 EXPERIMENTS

In this section, we evaluate the predictive performance and uncertainty quantification of the proposed models using several standard tasks including multivariate regression on UCI datasets, contextual bandits, and image classification, via comparing them with several well-established parameter-space variational inference approaches, i.e., weight-space BNN with KL divergence (KLBNN (Blundell et al., 2015)), and 2-Wasserstein distance (WBNN), and Bayesian deep ensemble (BDE), and function-space variational inference approaches, i.e., BNN with functional KL divergence (fBNN (Sun et al., 2019)), 2-Wasserstein FSVI (Rudner et al., 2022) and GWI (Wild et al., 2022).

Table 1: The table shows the results of average RMSE for multivariate regression on UCI datasets. We split each dataset randomly into 90% training data and 10% test data, and this process is repeated 10 times to ensure validity. We perform the paired-sample t-test for the results from our best one with other methods and get $p < .05$.

| Dataset | Yacht | Boston | Concrete | Kin8nm | Protein |
|---|---|---|---|---|---|
| ifBNN | $0.341 \pm 0.071$ | $0.607 \pm 0.093$ | $0.613 \pm 0.080$ | $0.758 \pm 0.048$ | $0.882 \pm 0.009$ |
| ifBDE | $\mathbf{0.089 \pm 0.096}$ | $\mathbf{0.393 \pm 0.087}$ | $\mathbf{0.324 \pm 0.079}$ | $\mathbf{0.334 \pm 0.034}$ | $\mathbf{0.779 \pm 0.010}$ |
| FSVI | $0.922 \pm 0.087$ | $0.967 \pm 0.095$ | $0.976 \pm 0.092$ | $0.992 \pm 0.043$ | $0.997 \pm 0.012$ |
| GWI | $2.198 \pm 0.083$ | $1.742 \pm 0.046$ | $1.297 \pm 0.053$ | $1.188 \pm 0.015$ | $1.333 \pm 0.007$ |
| fBNN | $1.523 \pm 0.075$ | $1.683 \pm 0.122$ | $1.274 \pm 0.049$ | $1.447 \pm 0.069$ | $1.503 \pm 0.025$ |
| WBBB | $2.328 \pm 0.091$ | $2.306 \pm 0.102$ | $2.131 \pm 0.068$ | $2.134 \pm 0.029$ | $2.188 \pm 0.012$ |
| KLBBB | $2.131 \pm 0.085$ | $1.919 \pm 0.074$ | $1.784 \pm 0.063$ | $1.787 \pm 0.027$ | $1.795 \pm 0.010$ |
| BDE | $0.210 \pm 0.003$ | $0.554 \pm 0.001$ | $0.543 \pm 0.003$ | $0.710 \pm 0.006$ | $0.873 \pm 0.001$ |

Table 2: The table shows the average test NLL on several UCI regression tasks. We split each dataset randomly into 90% of training and 10% of test. This process is repeated 10 times to ensure validity.

| Dataset | Yacht | Boston | Concrete | Kin8nm | Protein |
|---|---|---|---|---|---|
| ifBNN | $-1.347 \pm 0.95$ | $-0.316 \pm 0.30$ | $-1.366 \pm 0.20$ | $-2.410 \pm 0.14$ | $\mathbf{-1.942 \pm 0.31}$ |
| ifBDE | $\mathbf{-4.269 \pm 0.97}$ | $\mathbf{-2.406 \pm 0.75}$ | $\mathbf{-1.467 \pm 0.34}$ | $\mathbf{-3.051 \pm 0.28}$ | $-0.887 \pm 0.49$ |
| FSVI | $-0.650 \pm 0.87$ | $-0.487 \pm 0.52$ | $-0.720 \pm 0.33$ | $-1.774 \pm 0.08$ | $-1.692 \pm 0.58$ |
| GWI | $0.112 \pm 0.75$ | $-1.043 \pm 0.68$ | $-0.684 \pm 0.49$ | $-2.604 \pm 0.23$ | $-1.575 \pm 0.22$ |
| fBNN | $-0.770 \pm 0.86$ | $-1.193 \pm 0.76$ | $-1.001 \pm 0.52$ | $-2.445 \pm 0.62$ | $-1.486 \pm 0.23$ |
| WBBB | $2.856 \pm 0.18$ | $2.656 \pm 0.17$ | $2.838 \pm 0.15$ | $2.823 \pm 0.06$ | $2.744 \pm 0.02$ |
| KLBBB | $2.512 \pm 0.16$ | $2.066 \pm 0.11$ | $2.614 \pm 0.16$ | $2.614 \pm 0.07$ | $2.222 \pm 0.02$ |
| BDE | $8.085 \pm 2.13$ | $45.581 \pm 6.01$ | $63.367 \pm 22.05$ | $81.009 \pm 26.16$ | $62.305 \pm 53.15$ |

## 5.1 EXTRAPOLATION ILLUSTRATIVE EXAMPLES

Given a random polynomial function, some random data points are sampled as the observations. The setting details can be found in Appendix A3, and the results are shown in Figure 2. We can see that 1) BDE fitted the data points and recovered the underlying function well but exhibited an overfitting phenomenon (especially in the range [0.25, 0.75]); 2) ifBDE achieved the best performance among all methods and provided better uncertainty qualification than weight-space BDE; 3) both ifBNN and ifBDE have shown good performance on function fitting and, more importantly, well-calibrated uncertainty in the left-most and right-most ranges.

## 5.2 UCI REGRESSION

In this experiment, we evaluate our approaches for multivariate regression tasks on benchmark UCI datasets to demonstrate their performance on prediction and uncertainty estimation. The setting details can be found in the Appendix A3. Table 1 shows the predictive results evaluated by the root mean square error (RMSE). We can see that 1) Except for BDE, all functional inference approaches consistently provide better predictive results than the parameter-space ones, which shows the advantage of function-space variational inference.; 2) the functional one is better than its corresponding weight-space ones, such as ifBDE is better than BDE and ifBNN is better than WBBB; 3) BDE outperforms BNN in both weight-space and functional ones; 4) ifBDE is the best among all functional inference methods Furthermore, our proposed approaches significantly outperform all other two functional ones. Table 2 shows the average test negative log-likelihood (NLL) results, where the function-space ones are also better than the weight-space ones and ifBDE generally performs best among all function-space ones. We also see a significant performance drop in BDE.

## 5.3 IMAGE CLASSIFICATION AND OOD PREDICTION

We also evaluate our approaches to the image classification task with higher dimensions than UCI datasets. To demonstrate their performance on prediction and uncertainty estimation, we evaluate both the in-distribution and out-of-distribution (OOD) predictions on MNIST, FashionMNIST and

Table 3: Image classification and OOD detection performance.

| Model | MNIST Accuracy | MNIST OOD-AUC | FMNIST Accuracy | FMNIST OOD-AUC | CIFAR10 Accuracy | CIFAR10 OOD-AUC |
|---|---|---|---|---|---|---|
| ifBNN | $\mathbf{96.47} \pm 0.00$ | $\mathbf{0.92} \pm 0.01$ | $\mathbf{86.63} \pm 0.00$ | $0.84 \pm 0.01$ | $46.46 \pm 0.01$ | $\mathbf{0.70} \pm \mathbf{0.03}$ |
| ifBDE | $95.24 \pm 0.00$ | $0.85 \pm 0.03$ | $85.56 \pm 0.00$ | $\mathbf{0.88} \pm \mathbf{0.02}$ | $\mathbf{48.26} \pm \mathbf{0.01}$ | $0.62 \pm 0.03$ |
| FSVI | $96.26 \pm 0.00$ | $0.91 \pm 0.01$ | $85.28 \pm 0.00$ | $0.85 \pm 0.01$ | $46.46 \pm 0.01$ | $0.65 \pm 0.02$ |
| GWI | $95.40 \pm 0.00$ | $0.85 \pm 0.05$ | $85.43 \pm 0.00$ | $0.39 \pm 0.04$ | $44.78 \pm 0.01$ | $0.63 \pm 0.02$ |
| fBNN | $96.09 \pm 0.00$ | $0.80 \pm 0.07$ | $85.64 \pm 0.00$ | $0.81 \pm 0.02$ | $46.29 \pm 0.01$ | $0.61 \pm 0.03$ |
| WBBB | $96.16 \pm 0.00$ | $0.86 \pm 0.03$ | $85.57 \pm 0.00$ | $0.81 \pm 0.01$ | $45.76 \pm 0.01$ | $0.60 \pm 0.03$ |
| KLBBB | $96.26 \pm 0.00$ | $0.86 \pm 0.03$ | $85.71 \pm 0.00$ | $0.82 \pm 0.02$ | $46.20 \pm 0.00$ | $0.60 \pm 0.02$ |
| BDE | $95.32 \pm 0.00$ | $0.71 \pm 0.02$ | $69.95 \pm 0.07$ | $0.82 \pm 0.01$ | $24.47 \pm 0.01$ | $0.61 \pm 0.05$ |

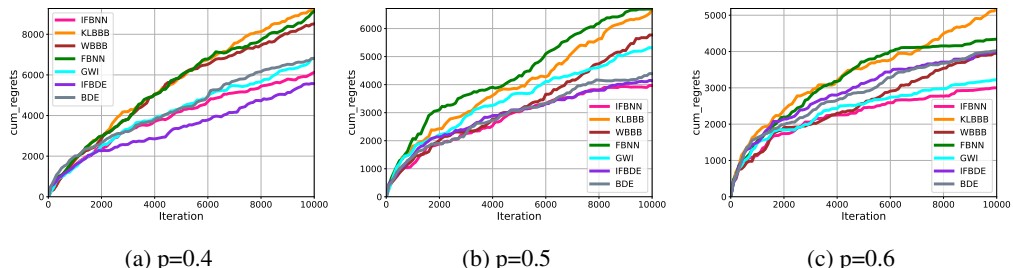

(a) p=0.4  (b) p=0.5  (c) p=0.6

Figure 3: Comparisons of cumulative regrets various methods on the Mushroom contextual bandit task (lower represents better performance).

CIFAR-10. Please see the Appendix for more setting details. The results are given in Table 3. We can see that 1) the weight-space approaches were similar or even slightly better than the traditional function-space ones; 2) our new approaches are better than others; and 3) ifBNN is slightly better than ifBDE.

## 5.4 CONTEXTUAL BANDIT

In this section, we evaluate the ability of ifBDL on contextual bandit problem following the setting in (Blundell et al., 2015) to guide exploration on the UCI Mushroom dataset fol, which includes 8124 instances, and each mushroom has 22 features and is identified as edible or poisonous. The agent can observe these mushroom features as the context and choose either to eat or reject a mushroom to maximize the reward. We consider three different reward patterns: for the action of eating a mushroom if the mushroom is edible, the agent will receive a reward of 5. Conversely, if the mushroom is poisonous, the agent will receive a reward of -35 with probabilities 0.4, 0.5, and 0.6 respectively for three different patterns, otherwise a reward of 5. On the other hand, if the agent decides to take the action of rejecting a mushroom, it will receive a reward of 0. The cumulative regrets of all parameter-space and function-space variational inference methods for 3 reward patterns are shown in Figure 3. We observe that 1) ifBDE and ifBNN achieve the best performance among all methods; 2) ifBDE is the best for 0.4 and 0.5, while ifBNN is the best for 0.6. It indicates that ifBDL can provide reliable uncertainty estimation in such decision-making scenarios.

## 6 CONCLUSION

In this paper, we propose two implicit functional BDL approaches: implicit functional Bayesian neural networks and implicit functional Bayesian deep ensembles, via the NTK-based GP transformation. These approaches leverage the tractable property of the 2-Wasserstein distance between Gaussian measures without sacrificing much on uncertainty modeling capability. These new BDL approaches demonstrate better predictive and uncertainty modeling capabilities compared to existing methods on benchmark tasks. Our future work will focus on improving the scalability of these approaches for large-scale models, such as GPT-4 and Gemini.

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

# A APPENDIX

## A.1 DERIVATION OF (14)

$$f_\infty(\mathbf{x}) = f_0(\mathbf{x}) - \Theta_{\mathbf{x}\mathbf{x}}\Theta_{\mathbf{X}}^{-1}\left(f_0(\mathbf{X}) - \mathbf{Y}\right)$$

and $\tilde{f}_t \sim GP\left(\bar{f}_t = f(\cdot; \boldsymbol{\mu}_t), \Lambda_{xx'}\right)$, where the mean function is easy to derive and we only show the derivation of covariance matrix below

$$\mathbb{COV}[f_\infty(X), f_\infty(X')]$$
$$= \mathbb{E}[f_\infty(X)f_\infty(X')^\top] - \mathbb{E}[f_\infty(X)]\mathbb{E}[f_\infty(X')]^\top$$
$$= \mathbb{E}[(f_0(X) - \Theta_{X\mathbf{x}}\Theta_{\mathbf{X}}^{-1}(f_0(\mathbf{X}) - \mathbf{Y}))(f_0(X') - \Theta_{X'\mathbf{x}}\Theta_{\mathbf{X}}^{-1}(f_0(\mathbf{X}) - \mathbf{Y}))^\top]$$
$$\quad - \mathbb{E}[f_0(X) - \Theta_{X\mathbf{x}}\Theta_{\mathbf{X}}^{-1}(f_0(\mathbf{X}) - \mathbf{Y})]\mathbb{E}[f_0(X') - \Theta_{X'\mathbf{x}}\Theta_{\mathbf{X}}^{-1}(f_0(\mathbf{X}) - \mathbf{Y})]^\top$$
$$= \mathbb{E}[f_0(X)f_0(X)^\top] - \mathbb{E}[f_0(X)\left(\Theta_{X\mathbf{x}}\Theta_{\mathbf{X}}^{-1}(f_0(\mathbf{X}) - \mathbf{Y})\right)^\top]$$
$$\quad - \mathbb{E}[\left(\Theta_{X\mathbf{x}}\Theta_{\mathbf{X}}^{-1}(f_0(\mathbf{X}) - \mathbf{Y})\right)f_0(X)^\top]$$
$$\quad + \mathbb{E}[\left(\Theta_{X\mathbf{x}}\Theta_{\mathbf{X}}^{-1}(f_0(\mathbf{X}) - \mathbf{Y})\right)\left(\Theta_{X\mathbf{x}}\Theta_{\mathbf{X}}^{-1}(f_0(\mathbf{X}) - \mathbf{Y})\right)^\top] - \mathbb{E}[f_0(X)]\mathbb{E}[f_0(X)^\top]$$

---

**Algorithm 1** ifBNN Inference

---

**Require:** Dataset: $\mathcal{D} = \{\mathbf{X}, \mathbf{Y}\}$, minibatch size B, loss function $\mathcal{L}$, BNN model $f_\theta: \mathbf{X} \to \mathbf{Y}$
1: Initialise $\theta = (\mu, \rho) \sim init(\cdot)$
2: **while** $\theta$ not converge **do**
3:      Sample $\mathcal{D}_B = \{\mathbf{X}_B, \mathbf{Y}_B\} \sim \mathcal{D}$                 ▷ fetch a data batch
4:      Sample $\mathbf{X}_{\mathcal{M}} \sim \mathbf{X} \,\&\, \mathcal{X}$                ▷ sample a measurement set
5:      Calculate data likelihood $\ell^{\text{lik}}$ using $\mathcal{D}_B$
6:      Evaluate network gradient $\nabla_w f(x)$ where $x \in \{\mathbf{X}_{\mathcal{M}}, \mathbf{X}\}$
7:      Evaluate covariance matrix $\Lambda$ and NTK $\Theta$   ▷ prepare Gaussian process on measurement set
8:      Calculate the Wasserstein regularizor, $\ell^{\text{w}}$ via (9)
9:      Optimise $\theta$: $\theta \leftarrow \theta - \alpha \left[ \nabla_\theta (\ell^{\text{lik}} + \beta \ell^{\text{w}}) \right]$
10: **end while**

---

$$
\begin{aligned}
&+ \mathbb{E}[f_0(X)]\mathbb{E}[\Theta_{X\mathbf{x}}\Theta_{\mathbf{X}}^{-1}(f_0(\mathbf{X}) - \mathbf{Y})]^\top + \mathbb{E}[\Theta_{X\mathbf{x}}\Theta_{\mathbf{X}}^{-1}(f_0(\mathbf{X}) - \mathbf{Y})]\mathbb{E}[f_0(X)]^\top \\
&- \mathbb{E}[\Theta_{X\mathbf{x}}\Theta_{\mathbf{X}}^{-1}(f_0(\mathbf{X}) - \mathbf{Y})]\mathbb{E}[\Theta_{X\mathbf{x}}\Theta_{\mathbf{X}}^{-1}(f_0(\mathbf{X}) - \mathbf{Y})]^\top \\
=& \mathbb{E}[f_0(X)f_0(X)^\top] - \mathbb{E}[f_0(X)]\mathbb{E}[f_0(X)^\top] \\
&- \mathbb{E}[f_0(X)\left(\Theta_{X\mathbf{x}}\Theta_{\mathbf{X}}^{-1}(f_0(\mathbf{X}) - \mathbf{Y})\right)^\top] - \mathbb{E}[\left(\Theta_{X\mathbf{x}}\Theta_{\mathbf{X}}^{-1}(f_0(\mathbf{X}) - \mathbf{Y})\right)f_0(X)^\top] \\
&+ \mathbb{E}[\Theta_{X\mathbf{x}}\Theta_{\mathbf{X}}^{-1}(f_0(\mathbf{X}) - \mathbf{Y})(f_0(\mathbf{X}) - \mathbf{Y})^\top \Theta_{\mathbf{X}}^{-\top}\Theta_{X\mathbf{X}}^\top] \\
&+ \bar{f}_t(X)\left(\Theta_{X\mathbf{x}}\Theta_{\mathbf{X}}^{-1}\left(\bar{f}_t(\mathbf{X}) - \mathbf{Y}\right)\right)^\top + \left(\Theta_{X\mathbf{x}}\Theta_{\mathbf{X}}^{-1}\left(\bar{f}_t(\mathbf{X}) - \mathbf{Y}\right)\right)\bar{f}_t(X)^\top \\
&- \Theta_{X\mathbf{x}}\Theta_{\mathbf{X}}^{-1}\left(\bar{f}_t(\mathbf{X}) - \mathbf{Y}\right)\left(\bar{f}_t(\mathbf{X}) - \mathbf{Y}\right)^\top \Theta_{\mathbf{X}}^{-\top}\Theta_{X\mathbf{X}}^\top \\
=& \mathbb{E}[f_0(X)f_0(X)^\top] - \mathbb{E}[f_0(X)]\mathbb{E}[f_0(X)^\top] \\
&- \mathbb{E}[f_0(X)\left(\Theta_{X\mathbf{x}}\Theta_{\mathbf{X}}^{-1}(f_0(\mathbf{X}) - \mathbf{Y})\right)^\top] - \mathbb{E}[\left(\Theta_{X\mathbf{x}}\Theta_{\mathbf{X}}^{-1}(f_0(\mathbf{X}) - \mathbf{Y})\right)f_0(X)^\top] \\
&+ \Theta_{X\mathbf{x}}\Theta_{\mathbf{X}}^{-1}\mathbb{E}[f_0(\mathbf{X})f_0(\mathbf{X})^\top]\Theta_{\mathbf{X}}^{-\top}\Theta_{X\mathbf{X}}^\top \\
&+ \bar{f}_t(X)\left(\Theta_{X\mathbf{x}}\Theta_{\mathbf{X}}^{-1}\left(\bar{f}_t(\mathbf{X}) - \mathbf{Y}\right)\right)^\top + \left(\Theta_{X\mathbf{x}}\Theta_{\mathbf{X}}^{-1}\left(\bar{f}_t(\mathbf{X}) - \mathbf{Y}\right)\right)\bar{f}_t(X)^\top \\
&- \Theta_{X\mathbf{x}}\Theta_{\mathbf{X}}^{-1}\mathbb{E}[f_0(\mathbf{X})]\mathbb{E}[f_0(\mathbf{X})^\top]\Theta_{\mathbf{X}}^{-\top}\Theta_{X\mathbf{X}}^\top \\
=& \mathbb{E}[f_0(X)f_0(X)^\top] - \mathbb{E}[f_0(X)]\mathbb{E}[f_0(X)^\top] \\
&+ \Theta_{X\mathbf{x}}\Theta_{\mathbf{X}}^{-1}\mathbb{E}[f_0(\mathbf{X})f_0(\mathbf{X})^\top]\Theta_{\mathbf{X}}^{-\top}\Theta_{X\mathbf{X}}^\top - \Theta_{X\mathbf{x}}\Theta_{\mathbf{X}}^{-1}\mathbb{E}[f_0(\mathbf{X})]\mathbb{E}[f_0(\mathbf{X})^\top]\Theta_{\mathbf{X}}^{-\top}\Theta_{X\mathbf{X}}^\top \\
&+ \mathbb{E}[f_0(X)]\mathbb{E}[f_0(\mathbf{X})^\top]\Theta_{\mathbf{X}}^{-\top}\Theta_{X\mathbf{X}}^\top - \mathbb{E}[f_0(X)f_0(\mathbf{X})^\top]\Theta_{\mathbf{X}}^{-\top}\Theta_{X\mathbf{X}}^\top \\
&+ \Theta_{X\mathbf{x}}\Theta_{\mathbf{X}}^{-1}\mathbb{E}[f_0(\mathbf{X})]\mathbb{E}[f_0(X)^\top] - \Theta_{X\mathbf{x}}\Theta_{\mathbf{X}}^{-1}\mathbb{E}[f_0(\mathbf{X})f_0(X)^\top] \\
=& \Lambda_{XX'} + \Theta_{X\mathbf{x}}\Theta_{\mathbf{X}}^{-1}\Lambda_{\mathbf{X}}\Theta_{\mathbf{X}}^{-\top}\Theta_{X'\mathbf{X}}^\top - \Lambda_{X\mathbf{x}}\Theta_{\mathbf{X}}^{-\top}\Theta_{X'\mathbf{X}}^\top - \Theta_{X\mathbf{x}}\Theta_{\mathbf{X}}^{-1}\Lambda_{\mathbf{X}X'}
\end{aligned}
$$

## A.2   ALGORITHMS

## A.3   EXPERIMENTAL SETTING

**Extrapolation illustrative examples.** We used an 1-D oscillation curve from the polynomial function: $y = \sin(3\pi x) + 0.3\cos(9\pi x) + 0.5\sin(7\pi x) + \epsilon$ with noise $\epsilon \sim \mathcal{N}(0, 0.5^2)$. There were 20 randomly sampled observation points, half of which were sampled from the interval $[-0.75, -0.25]$, and the other half are from $[0.25, 0.75]$. In this experiment, we used $2 \times 100$ fully connected tanh BNNs as variational posteriors for all models. The functional GP priors were pre-trained on the 20 training points for 100 epochs. We also used 40 inducing points for the sampling of marginal measurement points in FWBI, FBNN and GWI from $[-1, 1]$. All methods are trained for 10000 epochs.

**Multivariate regression on UCI datasets.** We choose BNNs posteriors with two hidden layers (input-10-10-output). The GP prior uses RBF kernel and is pre-trained on the test dataset for 100 epochs. The number of iterations for all models is 2000.

---

**Algorithm 2** ifBDE Inference

---

**Require:** Dataset: $\mathcal{D} = \{\mathbf{X}, \mathbf{Y}\}$, minibatch size B, ensemble size $M$, parameter initialisation scheme: $init(\cdot)$

1: Initialise $\{w_{z=1:Z}\} \sim init(\cdot)$
2: Sample $\mathcal{D}_B = \{\mathbf{X}_B, \mathbf{Y}_B\} \sim \mathcal{D}$                         ▷ fetch a data batch
3: Sample $\mathbf{X}_{\mathcal{M}} \in [\mathbf{X}, \mathcal{X}]$                   ▷ sample a measurement set
4: **while** $\{w_{z=1:Z}\}$ not converge **do**
5:     **for** $z = 1, ..., Z$ **do**
6:         Initialise $\widetilde{w_z} \sim init(\cdot)$ and denote $\widetilde{w_z} = \text{concat}(\{\widetilde{w_z}^{\leq L}, \widetilde{w_z}^{L+1}\})$
7:         Set $w_z^* = \text{concat}(\{\widetilde{w_z}^{\leq L}, 0^{L+1}\})$
8:     **end for**
9:     **for** $z = 1, ..., Z$ **do**
10:        Calculate data likelihood $\ell_z^{\text{lik}}$ using $\mathcal{D}_B$
11:        Evaluate network gradient $\nabla_w f_z(x)$ where $x \in [\mathbf{X}_B, \mathbf{X}_{\mathcal{M}}]$
12:        Evaluate covariance matrix $\Lambda_z$ and NTK $\Theta_z$
13:        Define $g_z(x) = \nabla_w f_z(x) w_z^*$
14:     **end for**
15: Calculate Wasserstein barycenter via (23)   ▷ prepare Gaussian process on measurement set
16: Calculate the Wasserstein regularizor, $\ell^{\text{w}}$ via (9)
17: Optimise $\theta$: $\theta \leftarrow \theta - \alpha \left[ \nabla_\theta (\sum_z \ell_z^{\text{lik}} + \beta \ell^{\text{w}}) \right]$
18: **end while**

---

**Classification and OOD detection** For all models in this experiment, the variational posteriors are fully connected BNNs with 2 hidden layers, each with 800 units. The functional prior is a Dirichlet-based GP designed for classification tasks and is pre-trained on test dataset for 500 epochs. ALL inference methods are trained for 600 epochs and the batchsize is 125.

**Contextual bandits.** The variational posteriors are fully connected tanh BNNs with two hidden layers (input-100-100-output) and the GP prior is pre-trained on 1000 randomly sampled points from training data. ALL models are trained using the last 4096 input-output tuples in the training buffer with a batch size of 64 and training frequency 64 for each iteration. All inference methods are trained for 10000 epochs.

A.4   Notation Table

Table 4 is the notation table to demonstrate the notation used in this paper.

Table 4: Notation table

| Notation | Meaning |
|---|---|
| $\mathcal{D} = \{(\mathbf{X}, \mathbf{Y})\} = \{(x_i, y_i)\}_{i=1}^{n}$ | a dataset with $n$ data points |
| $\mathcal{X} \subseteq \mathbb{R}^d$ | ($d$-dimensional) input space |
| $\mathcal{Y} \subseteq \mathbb{R}^c$ | ($c$-dimensional) output space |
| $\mathbf{X}$ | Finite marginal points |
| $\mathbf{X}_{\mathcal{M}}$ | Finite measurement points |
| $\mathbf{w}$ | Random model parameters for a BDL model (e.g., network weights of a BNN) |
| $f(\cdot; \mathbf{w})$ | Random function mapping defined by a BDL model parameterized by $\mathbf{w}$ |
| $\theta = \{\boldsymbol{\mu}, \boldsymbol{\sigma}\}$ | Parameters for variational distribution, $q_\theta(\mathbf{w}) = \mathcal{N}(\boldsymbol{\mu}, \boldsymbol{\sigma} \cdot I)$ |
| $p_0(\mathbf{w})$ | Prior distribution over model parameters (e.g., prior over weights in a BNN) |
| $p(\mathbf{w}|\mathcal{D})$ | Posterior over model parameters (e.g., posterior over weights in a BNN) |
| $q_\theta(\mathbf{w})$ | Variational posterior over model parameters (e.g., variational posterior over weights in a BNN) |
| $p_0(f)$ | Prior distribution over random functions |
| $p(f|\mathcal{D})$ | Posterior over functions |
| $q_\theta(f)$ | Variational posterior over functions |
| $\Theta$ | Neural tangent kernel (NTK) |
| $m$ | the mean function of a Gaussian process |
| $\mathbf{k}$ | the kernel function of a Gaussian process |
| $L$ | the layer number of deep learning |
| $M$ | the measurement set |

