# OpenReview forum: "Implicit Functional Bayesian Deep Learning"
_ICLR.cc/2025/Conference — Submitted to ICLR 2025_

### Official Review · Reviewer_fsyZ · 2024-10-25

**Soundness:** 2
**Presentation:** 2
**Contribution:** 2
**Rating:** 3
**Confidence:** 4

**Summary:**

In this paper, the authors propose a new objective for training variational Bayesian neural networks (VBNNs). Unlike the standard weight-space VBNN, the proposed ifVBNN uses the 2-Wasserstein distance between the implied posterior over functions $q(f \mid D)$ (as implied by the weight-space posterior $q(\theta \mid D)$) and a function-space prior (such as a GP).

The key of this approach is to approximate $q(f \mid D)$ with GP posterior resulting from the linearization of the network. Then, the W2 distance between this approximate functional posterior and the functional prior can be computed over a context set.

In addition to ifVBNN, the authors also propose ifBDE, which assumes that the weight-space posterior is given by a mixture of delta distributions. The implied GP is approximated through the Wasserstein barycenter of Gaussians.

**Strengths:**

* The paper is written clearly
* I believe the proposed method, while similar to previous approaches in generalized variational inference, is novel since ifBNN fixes the problems of previous works
* The experiments are quite diverse

**Weaknesses:**

* While the authors write the paper clearly, inaccuracies in their exposition hurt the overall presentation. For example, the authors confused "BNN" with "variational BNN". This is not a bad practice since it means the authors ignore other kinds of BNNs in the paper. For instance, the linearized Laplace approximation, which has a similar connection to the (empirical) NTK [1, 2, 3, etc.], is ignored. Moreover, the linearized Laplace has been extended to the function space [4], which achieves far better performance than the proposed approach.

* The proposed method seems very expensive since linearization must be done at every ELBO optimization iteration. This is in contrast to the functional Laplace [4], which only requires the Jacobian computation at test time. Indeed, during training, all one needs is MAP estimation. Similarly, in the VI literature, the author ignored [5], which I believe to be the state-of-the-art for function-space VI.

* Given the cost, the performance of the proposed method is sub-par. Even the deep ensemble version generalizes badly (95% acc on MNIST, 85% on FMNIST, and 48.26% on CIFAR-10). Moreover, the bandit experiment is very minimal with only a single dataset. Please see [6] for a better benchmark suite. Finally, the authors mentioned in their conclusion that their future work will be to scale the method to GPT-4 and Gemini. I strongly suggest the authors address the underperformance issues on small-scale datasets first.

* Better experimental settings should be used. I don't think MNIST, FMNIST, and other small scale datasets (unless they're for sequential decision-making) along with small-scale MLP models are useful anymore for truly evaluating the methods. In the age of large-scale models, the Bayesian deep learning community must adapt to the change. This means, studying the applications in LLMs, diffusion models, and other foundation models. This will force us, the community, to come up with BDL methods that are both theoretically sound _and_ computationally efficient. Example settings that the authors can use: [7, 8]

* The justification for the approximation of $q(f \mid D)$ from $q(w \mid D)$ via the NTK theory is shaky. It has been shown by [6] that even large models (which the authors didn't use) are not in the NTK territory yet --- using the NTK as a proxy is thus very shaky.


**References**

1. https://arxiv.org/abs/1906.01930
2. https://arxiv.org/abs/2008.08400
3. https://arxiv.org/abs/2306.03968
4. https://arxiv.org/abs/2407.13711
5. https://arxiv.org/abs/2312.17162
6. https://arxiv.org/abs/2310.00137
7. https://arxiv.org/abs/2308.13111
8. https://arxiv.org/abs/2402.05015

**Questions:**

1. Could the authors provide more details on whether the proposed loss is an ELBO, and thus, a valid VI objective?

2. What are the actual costs (wall clock and memory) of the method?

---

### Official Review · Reviewer_XLLu · 2024-10-27

**Soundness:** 3
**Presentation:** 3
**Contribution:** 2
**Rating:** 5
**Confidence:** 5

**Summary:**

Summary

- In this paper, the authors propose a new functional Bayesian Deep Learning (BDL) method, addressing the limitations of traditional BDL methods that rely on non-informative Gaussian priors in weight space. Specifically, they employ a function space prior and the 2-Wasserstein distance to improve prior specification. Utilizing findings from [1] and [2], they construct the function space posterior with a Gaussian Process and calculate the 2-Wasserstein distance using measurement points.

References

[1] Lee, J., Xiao, L., Schoenholz, S., Bahri, Y., Novak, R., Sohl-Dickstein, J., & Pennington, J. (2019). Wide neural networks of any depth evolve as linear models under gradient descent. Advances in neural information processing systems, 32.

[2] He, B., Lakshminarayanan, B., & Teh, Y. W. (2020). Bayesian deep ensembles via the neural tangent kernel. Advances in neural information processing systems, 33, 1010-1022.

**Strengths:**

Strengths

- The paper is well-written and easy to follow.
- Challenges with regularization using weight space priors have been consistently highlighted, and advancing BDL by proposing an improved method to address these issues is an important research direction.

**Weaknesses:**

Weaknesses

- Using the NTK to compute the covariance matrix requires the inversion of a matrix that scales quadratically with the size of the training dataset. This makes the computational cost prohibitively high, posing a challenge for applying this method to large datasets typical in the deep learning era.
- To my understanding, obtaining the Gaussian Process results in [1] and [2] requires a specific NTK initialization at the neural network's starting point to ensure convergence. This raises questions about how a functional posterior distribution can be applied using a general neural network, as convergence might not be guaranteed without this special initialization.
- The primary and critical point that prevents me from leaning towards acceptance is that the experiments were conducted only on a simple 2-layer MLP. Many methods exhibit significantly different behavior as model and data scales increase, and experiments on such a limited model scale do not provide sufficient evidence that the proposed prior distribution would be effective for training large-scale models.
- There is a lack of ablation studies such as the number of measurement points used and how performance changes as the number of these points increases. Including such ablation experiments would provide valuable insights.

References

[1] Lee, J., Xiao, L., Schoenholz, S., Bahri, Y., Novak, R., Sohl-Dickstein, J., & Pennington, J. (2019). Wide neural networks of any depth evolve as linear models under gradient descent. Advances in neural information processing systems, 32.

[2] He, B., Lakshminarayanan, B., & Teh, Y. W. (2020). Bayesian deep ensembles via the neural tangent kernel. Advances in neural information processing systems, 33, 1010-1022.

**Questions:**

See Weaknesses section.

---

### Official Review · Reviewer_Ks2n · 2024-11-04

**Soundness:** 2
**Presentation:** 4
**Contribution:** 4
**Rating:** 3
**Confidence:** 4

**Summary:**

In this work, the authors propose a novel variation of functional Bayesian neural networks in which they use the 2-Wasserstein measure instead of the commonly KL divergence when measuring the distance between the variationally posterior $q(f;w)$ to a GP prior $p_0(f)$. Although using the Wasserstein measure has been done before in the context of functional BNNs, it has not been done in the same methodological framework as here, where in order to evaluate the Wasserstein measure between $q$ and $p_0$, the authors leverage the neural-tangent kernel (NTK) theory to transform the BNN posterior to a Gaussian Process at every training step (after computing the likelihood), allowing computation of the Wasserstein measure - they call this implicit functional BNN (ifBNN). This allows BDL parametrization of the network, whilst allowing computing the Wasserstein measure. In addition, the authors propose implicit functional Bayesian Deep Ensemble (ifBDE). The core idea in ifBDE is to train multiple neural networks, but constraining the aggregated function of the ensemble via regularisation by usage of the Wasserstein measure between the aggregate ensemble function $q(f;w)$ and  $p_0(f)$. The authors then showcase and benchmark their method in regression, classification and a contextual bandit setting.

**Strengths:**

* The paper reads extremely well. The content is clear and it is generally a pleasure to read.
* The ifBNN method is well motivated, and although simple, in my opinion elegant and novel.
* Functional BNN methods are becoming increasingly popular due to issues with weight space priors, and is also highly relevant due to the increasing focus on uncertainty quantification capabilities of neural networks.

**Weaknesses:**

I here list my weaknesses, but will elaborate on them in questions.

1. I generally find the experimental section highly lacking and some of the results have me a little concerned about implementation details. This is my primary reason for the low score, which if addressed, I would be happy to increase my score.
2. The motivation for the implicit functional BDE is not clear to me.
3. There are little-to-no details provided required for reproducibility.

**Questions:**

To elaborate:

* With regards to experiments:
     1. Could the authors provide NLL metrics on the classification tasks, and perhaps also the commonly used ECE? There are currently no in-distribution uncertainty quantification evaluation on the classification datasets, which in my opinion are highly important in these settings.
     2. I am a bit concerned/confused by the BDE results on all of the presented problems. If I understand correctly: BDE is the ensembling of KLBBB models, and with that in mind, it is not clear to me how it is possible that BDEs can be performing worse than individual KLBBB models?
     3. Although the authors compare to several methods, I think there is a large class of highly popular BNN methods missing such as Last Layer Laplace methods and simply regular deep ensembles. This is especially important due to the fact that weight space variationally inference BNNs are known to be quite brittle. Unfortunately I do not think that the paper currently shows the full picture of comparable methods.
* Could the authors comment a bit on the motivation behind implicit functional BDE? Moreover, could the authors comment on whether or not they have observed if the functions that stem from the members in the ifBDE models are functionally diverse? It is not clear to me that the specification the authors have of these models would not simply cause the models to collapse to the same functions since they are jointly trained.
* Do the authors have some references for their specification of BDEs? I have not seen this particular specification of them before, and there does not appear to be any reference on this specification.
* Could the authors comment on scalability issues of these methods? Do they suffer from the same scalability issues as other NTK methods?
* Although there are some details presented on network and method hyperparameters, the majority of details are missing such as prior specification and tuning in the compared methods, optimizer details, network activation functions are missing, just to list a few. Good choices of these are key for these methods to work, and without at least a few listed, it is hard to say if these are confounders in the results.

Minor comment:
* In line 359 it is not clear to me what "... can achieve accuracy of $10^{-10}$ ..." means.

---

### Official Review · Reviewer_UU9j · 2024-11-05

**Soundness:** 3
**Presentation:** 2
**Contribution:** 3
**Rating:** 5
**Confidence:** 4

**Summary:**

This paper considers the common problems of Bayesian Deep Learning when using the KL divergence. To address these issues, they propose to switch from weight-space prior (and KL) into functional inference (with Wasserstein distance), which is recently a common research direction. To increase the predictive and uncertainty estimation capabilities while using Wasserstein distance, they propose two novel implicit functional approaches. The main idea is to implicitly transform the posterior of the Bayesian Deep Learning method to a GP via NTK and get tractable 2-Wasserstein distance computation. In particular, the authors consider two Bayesian Deep Learning methods — Bayesian Deep Ensemble and BNNs. Finally, they test these approaches on a set of experimental settings, achieving promising results.

**Strengths:**

The paper has a few significant strengths, which I will outline below.


**Strengths:**
1. Proposing a novel method in the direction of functional-space priors in Bayesian Deep Learning, which might be useful for easier Wasserstein distance computations, and, as such, increasing the models flexibility and uncertainty estimations.
2. In fact, the authors show how to incorporate their main idea with two popular Bayesian Deep Learning methods (i.e., BNNs and Bayesian Deep Ensemble) arising two novel implicit functional approaches.
3. The proposed approaches were checked on a set of experimental settings, achieving promising results — usually better than other functional-space and weight-space methods.

**Weaknesses:**

Despite its strengths, the paper has a few major and minor weaknesses.

**Major weaknesses:**

1.  Lack of comparison with the latest methods in the functional-space research direction - e.g., [1], [2], [3], and [4].
2. More visual examples (e.g., 2d classification) would be needed to better understand how good are these methods in uncertainty quantification? Maybe worth to consider the examples from [5].
3. Missing comparison against different sizes of BNNs - I would like to understand what are the current computational limits of these approaches.
4. I think that this paper is missing some justifications and introducing notations, e.g., in 3.2, I don't know where $L$ came from and assume it's from [6]. Overall, the 3.1 and 3.2 should be rewritten in a more approachable way.
5. Another issue - I'm not sure about the presented results, when I checked (e.g., UCI regression) in GWI paper, the results for most of the methods were much different from the one presented in this paper. Could you explain why?
6. Finally, the GWI paper uses tanh activation function and you are using rbf, why? Is it a source of the different results?



**Minor weaknesses:**

1. In Theorem 1, the notation for $f_{in}$, $m_{in}$ and $\Sigma_{in}$ might be hard to follow at the first glance. So, I will suggest to slightly change it.
2. In line 309, should be "Wasserstein"
3. In Algorithm 1 and Algorithm 2, we should have "training," not "inference," right?


**References:**

[1] Rudner, T. G., Kapoor, S., Qiu, S., & Wilson, A. G. (2023, July). Function-space regularization in neural networks: A probabilistic perspective. In International Conference on Machine Learning (pp. 29275-29290). PMLR.

[2] Cinquin, T., Pförtner, M., Fortuin, V., Hennig, P., & Bamler, R. (2024). FSP-Laplace: Function-Space Priors for the Laplace Approximation in Bayesian Deep Learning. arXiv preprint arXiv:2407.13711.

[3] Wu, M., Xuan, J., & Lu, J. Functional Wasserstein Bridge Inference for Bayesian Deep Learning. In The 40th Conference on Uncertainty in Artificial Intelligence.

[4] Haddouche, M., & Guedj, B. (2023). Wasserstein PAC-Bayes Learning: Exploiting Optimisation Guarantees to Explain Generalisation. arXiv preprint arXiv:2304.07048.

[5] Tran, B. H., Rossi, S., Milios, D., & Filippone, M. (2022). All you need is a good functional prior for Bayesian deep learning. Journal of Machine Learning Research, 23(74), 1-56.

[6] Bobby He, Balaji Lakshminarayanan, and Yee Whye Teh. Bayesian deep ensembles via the neural tangent kernel. Advances in Neural Information Processing Systems, 33, 2020.

**Questions:**

I’m considering if there is anyway to incorporate the Wasserstein Gradient Flow (being sometimes used in generative modeling) into this field of functional-space priors?

---

### Meta-Review · Area_Chair_1wqS · 2024-12-18

**Metareview:**

The paper proposes a method for Bayesian deep learning that moves from KL minimization to Wasserstein distance as measure. The four reviewers of the paper were in agreement that the paper was not good enough for acceptance to ICLR. There was no rebuttal or discussion. The major concern raised was in insufficient experimental validation of the method.

**Additional Comments On Reviewer Discussion:**

The authors didn't provide a response to comments, so there was no discussion.

---

### Decision · Program_Chairs · 2025-01-22

Reject